# The Slight Adjustment in the Weight of Sulfur Sheets to Synthesize β-NiS Nanobelts for Maintaining Detection of Lower Concentrations of Glucose through a Long-Term Storage Test

**DOI:** 10.3390/nano13162371

**Published:** 2023-08-18

**Authors:** Hsiensheng Lin, Chengming Peng, Jenbin Shi, Bochi Zheng, Hsuanwei Lee, Pofeng Wu, Minway Lee

**Affiliations:** 1Department of Electronic Engineering, Lunghwa University of Science and Technology, No. 300, Sec. 1, Wanshou Rd., Guishan, Taoyuan 333326, Taiwan; shemglin@ms63.hinet.net; 2Department of Medicine, Chung Shan Medical University, No. 110, Sec. 1, Chien-Kuo N. Rd., Taichung 402367, Taiwan; 3Division of General Surgery, Department of Surgery, Chung Shan Medical University Hospital, No. 110, Sec. 1, Chien-Kuo N. Rd., Taichung 402367, Taiwan; 4Da Vinci Minimally Invasive Surgery Center, Chung Shan Medical University Hospital, No. 110, Sec. 1, Chien-Kuo N. Rd., Taichung 402367, Taiwan; 5Department of Electronic Engineering, Feng Chia University, 100, Wen-Hwa Rd., Seatwen, Taichung 407102, Taiwan; 6Ph.D. Program of Electrical and Communications Engineering, Feng Chia University, 100, Wen-Hwa Rd, Seatwen, Taichung 407102, Taiwan; bochi0020@gmail.com (B.Z.); rocklee3001g@gmail.com (H.L.); 7Department of Electrophysics, National Chiayi University, Chiayi City 60004, Taiwan; gavinwu0325@gmail.com; 8Department of Physics, Institute of Nanoscience, National Chung Hsing University, 250 Kuo Kuang Rd., Taichung 40227, Taiwan; mwl@phys.nchu.edu.tw

**Keywords:** electrodeposition, β-NiS, nanobelt, non-enzymatic, glucose sensor

## Abstract

The β-nickel sulfide (β-NiS) nanobelts were fabricated by electrodepositing a nickel nanosheet film on Indium tin oxide (ITO)-coated glass substrates and sulfuring the nickel film on ITO-coated glass substrates. The sulfurization method can be used to form nanobelts without a template. A small glass tube was used to anneal the sulfur sheet with a nickel nanosheet film. After applying vacuum to the tube, the specimen was annealed at 500 °C. By adjusting the weight of the sulfur sheet in a small glass tube, a nanobelt structure can be formed on the film for 4 h. The β-NiS nanobelt film had a sulfide and nickel molar ratio that was nearly 0.7 (S/Ni). After five years of a long-term storage test, the β-NiS nanobelt films were able to measure the glucose in a solution with the value of sensitivity of 8.67 µA cm^−2^ µM^−1^. The β-NiS nanobelt film also detected glucose with a limit of low detection (LOD) of around 0.173 µM. The estimation of reproducibility was over 98%. Therefore, the β-NiS nanobelt film has a significant ability to detect low concentrations of glucose in a solution.

## 1. Introduction

NiS has been employed in many applications, as super-capacitors and a co-catalyst for hydrogen evolution reaction, and as an electrode material of lithium rechargeable batteries in the past decades [1,2,3,4,5,6,7]. In addition, some researchers also performed this material that had a chance in applying biosensor to detect the glucose concentration [7]. The use of NiS material in these applications has intriguing potential regarding the non-enzymatic glucose sensors. Therefore, other researchers have studied and developed the electrodes of the NiS material to detect the glucose in a solution [8,9,10]. The structure of the non-enzymatic glucose sensors is not complicated. The NiS film on a small electrode is the only thing visible. It has a few advantages over other enzymatic glucose sensors. The sensor is resistant to changes in temperature and humidity [8,9,10,11,12].

In general, many different methods of glucose detection have been developed by using high-performance liquid chromatography, capillary zone electrophoresis, glucose oxidase, copper iodometry, and non-enzymatic glucose sensors [13,14]. The detection methods for sensing glucose can be utilized in various applications, including bio-industrial processes, the food industry, and medical research [13,14,15,16,17,18,19]. However, high-performance liquid chromatography and capillary zone electrophoresis methods were more complex. The sensor with glucose oxidase was not preserved over many years at room temperature easily. Some of the applications used the non-enzymatic glucose sensors for medical monitors and commercial products in biosensors because the non-enzymatic glucose sensors were built quickly and developed the electrodes more easily against temperature and humidity than other sensors with glucose oxidase [14,15,16,17,18,19,20]. It is also believed that the detection of a low glucose concentration is a necessary of cell culture in the medical research [21,22,23]. Some researchers have cultured cells in solution with low levels of glucose [23]. For the diabetic retinopathy (DR) study, researchers (Yuanping et al., 2018) implemented the cell cultures in different low glucose concentrations in a solution. The result showed that the low glucose (15 µM) on retinal pigment epithelium (RPE) cells was suitable for inducing reactive oxygen species (ROS) production and cell mitophagy. The researchers also measured the various glucose consecrations to uptake glucose from the measurement of cultured cells or cell wall yeast by using a NaOH solution [24,25,26]. Therefore, it could be necessary to develop biosensors for sensing glucose at a relatively low concentration. A number of literature types were reviewed on the use of NiS film on the electrode for sensing glucose at a relatively low concentration [8,9,10]. Many different materials for the non-enzymatic glucose sensors were reported by researchers [27].

Some researchers (Zhijing Yu et al., 2016) proposed the NiS nanobelt for the application of the rechargeable batteries at the cathode electrode. An ion (Ai^3+^) of the rechargeable was taking advantage of the nanoscale structure for enhancing ion diffusion and cathodes measured the low overpotential, the high storage capacity, and good cyclability [28]. Researchers (Xiaoyan Zhuo et al., 2021) also reported that the heterostructure of the nanobelt array between Ni_3_S_2_ and Ni could be developed into an inexpensive method for hydrogen production and urea wastewater [29]. Researchers also introduced the structure of the nanobelt array, which increased the reaction for hydrogen production and urea wastewater. However, the electrode of β-NiS nanobelts was little covered in the report for sensing the glucose [7,8,9,10,11,12,13,14,15,16,17,18,19,20,27]. The β-NiS nanobelt electrode was also reported less in a long storage trial, and the specimens were conducted in a storage test for more than five years.

The ability of the β-NiS nanobelt film on the electrode was tested to detect the different glucose concentrations in a solution after a long-term storage test in this study. Therefore, the synthesis and analysis of β-NiS films are research on the specific topic. The development of β-NiS nanobelt film can be achieved through an inexpensive electrodepositing method and a safe sulfurization process for synthesizing β-NiS nanobelt film in this study [30,31,32,33]. The crystallography of the β-NiS nanobelts was investigated on the glass substrate using X-ray diffraction (XRD). Field emission scanning electron microscopy (FE-SEM) was used to investigate the morphologies of the β-NiS nanobelts. Furthermore, the β-NiS nanobelt films were preserved for more than five years and analyzed their characteristics after excellent preservation at room temperature. In this report, the ability of the β-NiS nanobelt film to detect low glucose levels in a solution was analyzed.

## 2. Materials and Methods

### 2.1. Materials and Reagents

The fabrication of a β-NiS film was carried out through electrodepositing film and the sulfurization process. First, the nickel film was manufactured by using electroplating. The nickel film was created on ITO glass substrates (cut into 0.5 cm × 2 cm specimens) using an electrodeposition method. The containing electrolyte was 0.1 M NiSO_4_·6H_2_O with 98.5%, including 0.05 M NaOH with 96%. The pharmaceutical manufacturers were Sigma-Aldrich and Showa individually. The electrodeposition process was set at 3.0 V DC. The temperature on the electrolyte was set at 60 °C. The electrodeposition process was set for 10 min in a solution of pH 7.7 solution [30,31]. In the sulfurization process, a physical vapor transport (PVT) process was utilized. In different vacuum-sealed glass tubes, the sulfur sheets (0.33–0.50 mg) were individually annealed with the Nickel films. The nickel sulfide films were annealed for 4 h at 500 °C [32,33].

### 2.2. Surface Morphology and Optical Characterization

The particle size of the specimens was analyzed by using FE-SEM. The instrument was an energy dispersive spectrometer (EDS). The model number of the instrument with EDS analysis is HITACHI S-4800 (Hitachi, Tokyo, Japain). The instrument of XRD was ready for analyzing the crystallography on the specimens. The model number of the instrument for XRD analysis is SHIMADZU XRD-6000. In this work, an ultraviolet/visible/near-infrared (UV/Visible/NIR) spectrometer was also used to survey the β-NiS particles with absorption spectra. The model number of the instrument for optical spectrum analysis is Hitachi U-3501. Finally, an instrument with the high-resolution transmission electron microscopy (HR-TEM) was used to analyze β-NiS nanobelts. The model number of the equipment is JEOL JEM-2010.

### 2.3. Electrochemical Measurements

The measurement of current responses was also implemented on the β-NiS films by using amperometry. Cyclic voltammetry (CV) was utilized to measure cyclic voltammogram. For the reference electrode, an electrode with Ag/AgCl was utilized. A potentiostat (CH Instruments, CHI-700 and CHI-1000, CH Instruments, Bee Cave, TX, USA) was utilized for the measurement. The Pt electrode was used in the three-electrode configuration for the β-NiS films. The CV measurements of the β-NiS films were conducted in a 0.1 M NaOH solution. The values of the measurement were recorded regarding the current responses for the variable glucose concentrations. The pharmaceutical manufacturer of the glucose (≥99.5%) was Sigma-Aldrich. Measurement of current responses was also performed on the β-NiS films using amperometry.

## 3. Results and Discussion

### 3.1. Characterization of Ni Nanomaterial

With regard to fabrication of the β-NiS film, the nickel films were first synthesized by using electrodeposition. The nickel film was observed with many nickel nanosheets, as shown in Figure 1. The cross-sectional morphology of the nickel film is shown in Figure 1.

The image shows a formed structure of the nanosheets on the nickel film, which was estimated to have a particle size of 0.03–0.2 μm in the inset of Figure 1.

### 3.2. XRD Analysis of Ni Films

Figure 2 displays the XRD pattern. The XRD profiles of the nickel film were verified by applying the Joint of Committee on Powder Diffraction Standards JCPDS card (JCPDS870712) [9]. The PVT method was used for synthesizing the NiS film.

### 3.3. XRD Analysis of β-NiS Films

Table 1 presents the results for the NiS film, which were obtained by annealing sulfur sheets of different weights with the Ni film. This table shows that the different weights of the sulfur sheets ranged from 0.33 mg to 0.5 mg. The NiS film was synthesized by annealing for 4 h at 500 °C with the nickel film and sulfur sheets of different weights.

The result of the pattern was drawn to illustrate the distribution weight using sulfur sheets of different weights for the end products. As seen in Figure 3, the permutation of the NiS composite material was arranged from top to bottom with respect to Ni_3_S_2_, β-NiS, and α-NiS materials. The lighter sulfide sheet was fabricated with the nickel film, and it was used for synthesizing the NiS material. It exhibited more potential for forming β-NiS nanobelts.

The analysis was the main focus of the β-NiS films, which presented intriguing results. First, the β-NiS films were analyzed by applying XRD. Figure 4 shows the peak patterns of β-NiS films by applying XRD. They were confirmed by the JCPDS card (120041).

### 3.4. FESEM-EDAX Analysis of β-NiS Films

Figure 5 also displays the FE-SEM image of the specimen for the β-NiS film. The sulfur sheets weighing 0.36 mg were annealed with the nickel film for 4 h at 500 °C.

A FE-SEM image is shown as a nanobelt structure in Figure 5. The β-NiS nanobelts were observed that were presented to point many different directions on the β-NiS film. The weight of the sulfur sheet was the value of 0.36 mg. It was with the nickel film by synthesizing the β-NiS nanobelt film. The β-NiS nanobelts showed an average length of approximately 1.5 μm. Some irregular particles were also observed on the β-NiS film. Figure 5 displays a particle size of 0.5–2 μm.

The sulfur sheets weighing 0.43 mg were used to produce the β-NiS film. The image shows many irregular particles for the β-NiS material in Figure 6. The inset of image also shows a few irregular particles which are the size of 0.5–3 μm on the film in Figure 6.

The specimens of the β-NiS films were analyzed by applying FE-SEM with an EDS. The EDS results for the β-NiS films are shown in Figure 7. The top pattern in Figure 7 shows the weight percentage of the final compound with 72.85 wt.% and 27.15 wt.% for the β-NiS nanobelts. The calculation result of a molar ratio is the value of approximately 0.7 (S/Ni).

The β-NiS irregular particles with weight percentages of 70.57 wt.% and 29.43 wt.% are presented in the image of the bottom pattern in Figure 7. The calculation result of a molar ratio is approximately the value of 0.8 (S/Ni). With respect to the results of EDS analysis patterns for the β-NiS films, these were like the other EDS analysis patterns [34]. The final products of the β-NiS films presented an important result. The adjustment of the weights for the sulfur sheet is an intriguing method. It was shown that a lighter sulfide sheet combined with the nickel film could present a chance to create the β-NiS nanobelts.

With regard to growth mechanism for the structure of the β-NiS, researchers (Denholme et al., 2010) reported that NiS_2_ can be fabricated in the Ni–S system by using the surface-directed vapor transport method [32]. Researchers (Denholme et al., 2012) also reported that TiS_2_ and TiS_3_ were synthesized in the Ti–S system by using a PVT method [33]. The researchers adjusted the temperature to fabricate the different particle sizes and the morphology in a Ni–S system. They also used the surface-assisted chemical vapor transport (SACVT) methods to increase the appropriate sulfur powder to fabricate the different structures with the flower-like or nanoribbons of TiS_x_ (x = 2 or 3) [32,33]. The researchers adjusted the different weights of the reactant power in the Ti–S system to control the S vapor pressure. The TiS_3_ films have been altered by growth kinetics. The reaction of the final product can be observed. The reaction was reliant on the variable vapor pressure. It could be considered that the morphology in the synthesizing process of the NiS material changed as the S vapor pressure reduced or increased obviously, as in the synthesizing process of the TiS_x_ material [9,32,33]. Thus, it can be concluded that it was possible to synthesize β-NiS nanobelts in a vacuum system by adjusting the weight of the sulfur sheet for producing the different vapor pressures.

In this work we decided to study more measurements for the β-NiS nanobelts because it could induce the interest of researchers more than the irregular particles. The β-NiS nanobelt films were analyzed by using FE-SEM again. The FE-SEM image is shown in Figure 8. From this image, it can be observed that the nanobelts were approximately 50–500 nm in width and that average length was the value of approximately 1.5 μm.

The β-NiS films carried out a test for product life cycle. The specimens were kept in small containers with covers. These specimens were placed on a table in the laboratory for more than five years. Preserving the β-NiS nanobelt films for an unusually long time was conducted in the test from 2011 to 2016 in a laboratory. Previous literature (Lin et al., 2018) reported the conditions of the test for the product life cycle to be as follows: temperatures of 16–26 °C and relative humidity of 50–65% in the laboratory [9]. After completing the test for product life cycle over five years, the specimens were measured using optical and electrochemical instruments.

### 3.5. Optical Analysis of β-NiS Films

The optical instrument was used to analyze the specimen of UV/Visible/NIR absorption. An optical spectrometer could be used to record the results of the β-NiS material in the optical spectrum with a range from 300 to 1200 nm. For analyzing the β-NiS nanobelts and the irregular particles, they were taken from the glass substrate in a solution of water for the optical spectrometer. The measurements of the optical spectrums are shown in Figure 9.

To determine the energy gap of the nanobelts and irregular particles, it can be used as in Equation (1) [35]:*αhν* = *C*(*hν* − *E_g_*)*^m^*(1)
where *C* denotes the fixed and separate values. The symbol *hν* represents the photon energy. The energy gap is represented by the symbol *E_g_*. The *m* signifies the value of 0.5. The inset of the image shows the values of the measurements of (*αhν*)^2^ against *hν* in Figure 9. Plotting the tangent lines of the two curves show the two dotted lines to estimate the two values of the energy gap. The energy gap of the β-NiS irregular particles was estimated to be 0.6 eV. The other energy gap of the specimen for the β-NiS nanobelts with the irregular particles was checked with the value from 0.6 eV to 1.2 eV. The energy gap of β-NiS nanobelts was observed with an increase in energy gap. The energy gap of the β-NiS nanobelts was determined to be approximately 1.2 eV. The quantum confinement was considered for this effect [36,37].

### 3.6. Electrochemical Analysis of β-NiS Films

Upon the completion of the long-term storage test, the electrochemical characteristics of the β-NiS nanobelt films can be realized using CV and amperometry. The parameters for sensing glucose could be revealed, including sensitivity, selectivity, stability, and reproducibility. The values of measurement were recorded by CV in 0.1 M NaOH for the present current responses from the variable glucose concentrations. In Figure 10a,b, the cyclic voltammograms (CVs) of the β-NiS films are presented. At a rate value of 20 mVs^−1^, the electrochemical instrument was set to scan.

The detection area on the specimen was 0.5 cm × 1 cm, and it was utilized to detect the variable glucose concentrations. The specimens with a potential range of 0–0.8 V were measured using CV measurements for one cycle. By drawing the red curve in the inset of Figure 10a, the CV result for the bare ITO has been presented. The red CV curve corresponds to the efficient measurement of the bare ITO glass substrate. With the particles of the different sizes on its surface, the blue CV curve indicates the performance of the β-NiS film. A redox peak was not observed at the potential at 0.6–0.7 V, as shown by the blue CV curve in a cyclic voltammogram. In a NaOH solution, the β-NiS nanobelt film was also measured. The test condition of NaOH was also set at 0.1 M. As illustrated in Figure 10b, the variable glucose concentrations (0 µM, 10 µM, 20 µM, 30 µM, and 35 µM) can be measured. The results of the measurement of the redox curves of nano-NiS materials have been reported in some literature. In some literature reports, it has been suggested that Equation (2) was proposed to describe the electrode of the NiS material in a NaOH solution [7,8,9,10,38,39,40,41,42,43]:Ni^2+^ → Ni^3+^ + e^−^(2)

According to the researchers, the NiS material electrode was followed by a reduction-oxidation reaction to detect glucose from the redox Equation (3) in a NaOH solution [8,9,10,42,43]:Ni^3+^ + glucose → Ni^2+^ + gluconolactone(3)

From the above two equations (Equations (2) and (3)), the β-NiS nanobelt film followed the two reduction-oxidation equations for glucose detection. The inset of Figure 10b displays the values of the current responses against different glucose concentrations. The values of the oxidation peaks have been indicated for the current responses. To estimate the glucose detection range in the inset of Figure 10b, a red line was drawn. The red line’s coefficient of determination was estimated around the value of ≈0.99 (R^2^ = 0.9898). The result of the glucose detection range could be accepted with approximately 1–35 μM.

The amperometric measurement was performed using two β-NiS nanobelt films. The serial numbers of the specimens for the different β-NiS nanobelt films were No. 3 and No. 10. Amperometry was used to measure the variable glucose concentrations of the β-NiS nanobelt material with a range of 1–45 μM. The values of the current responses from the variable glucose concentrations can be seen in Figure 11 and Table 2.

Figure 11 shows that the values of the current response to the varying glucose concentrations were observed from 1 µM to 45 µM. In the inset (top right) of Figure 11, the average values of current response were also recorded from testing the specimen (No. 3). The results of measurement could be drawn as a red trendline with a linear relationship. The red trendline was shown with a correlation coefficient of 0.9928 in the inset (top right) of Figure 11 for the specimen No. 3.

This could be described by Equation (4).
*I* [m Acm^−2^] = *ζ*_slop_ × C_glucose_ [µM] + 0.28 ± 0.03(4)
where the *ζ*_slop_ is the value of sensitivity for the equation. The *ζ*_slop_ is the value of 8.67 ± 0.76 µA cm^−2^ µM^−1^. The intercept in Equation (4) has been estimated around 0.28 ± 0.03. The β-NiS nanobelt film was also ready for selectivity measurement. The result for the selectivity of the β-NiS nanobelt electrode is shown in the inset (top left) of Figure 11. The relative response from the various interferences has been recorded with the different addition. The interference effects were minimal with 2 µM uric acid (UA), ascorbic acid (AA), and 4-acetamidophenol (AP) during the detection of the glucose. The measurements for a duration of 7 days were carried out by using the two specimens (No. 3 and No. 10). The values of the measurement for a period of 7 days were also recorded in Figure 11 and Table 2. The values of the current response using the specimen (No. 10) decreased by 76% compared to the initial value on the seventh day. However, the values of the current response using the specimen (No. 3) only decreased by 10% from the initial value on the seventh day. The current response of the β-NiS nanobelt film was measured by using the fixed glucose concertation. Ten successive amperometric curves of 10 μM glucose using the β-NiS nanobelt film were recorded by amperometry, as shown in Figure 12.

The values of the β-NiS nanobelt film can be calculated using the average of the current response, standard deviation (SD), relative standard deviation (RSD), and reproducibility (see Table 2). Two types of instruments were used to measure the current responses (CH Instruments, CHI-700 and CHI-1000)). One of the instruments was carried out the measurement for ten successive amperometric curves of 10 μM glucose using the β-NiS nanobelt film (No. 3). The other instrument also carried out the measurement for ten successive amperometric curves of 10 μM glucose using the β-NiS nanobelt film (No. 3). There were two specimens (No. 3 and No. 10) for measuring the values of current response over twenty times. The values of measurement using the specimen (No. 10) were also recorded in Table 2. The result of the reproducibility could be estimated by using the recorded values in Table 2. The average RSD values of 0.91% (No. 3) or 1.79% (No. 10) were significantly less than 10% and the reproducibility of 99.66% (No. 3) and 98.88% (No. 10) satisfied the general measurement system quality [44].

The stability of the measurement was carried out at 10 μM glucose by using specimens (No. 3) over 30 min. The current response of the specimen decreased around 2.6% (from 0.3381 to 0.3293 mA) after 30 min (Figure 12, insert). The deviation obtained is less than 10% of the average current response. Thus, the specimen (No. 3) shows enough stability of response for glucose sensing.

The specimen was also measured for the low concentration detection by using the β-NiS nanobelt film for three times in 0.1 M NaOH. The LOD value of nearly 0.1 µM was confirmed by using hand tests (see the inset at the bottom of Figure 11). However, the LOD formulae can be written down as Equation (5) [45]:LOD = 3 S_b_/m(5)

The S_b_ is the standard deviation. Based on the measurement (0.0005 mA) in the blank signal, it can be calculated that the LOD is in the value of approximately 0.173 µM. The limit of quantification (LOQ) can be described in the following Equation (6) [45]:LOQ = 10 S_b_/m(6)
where the m is slop value of 8.67 µA µM^−1^. The result of the computation is the value of 0.577 µM. Finally, Table 3 presents a comparison of the glucose detection performance of the nickel sulfides reported in some previous literature.

### 3.7. HR-TEM Analysis of β-NiS Nanobelts

The HR-TEM instrument was utilized to analyze the β-NiS nanobelt film. The HR-TEM images can be seen in Figure 13.

After they were synthesized, the β-NiS nanobelts were analyzed by using an HR-TEM analysis. The image of a broken nanobelt used in the analysis of the lattice fringes with A and an individual nanobelt can be seen in the inset of Figure 13a. The lattice fringes in Figure 13b are visible on the HR-TEM image with an interspace of 0.2772 nm. The distance between two adjoining planes (300) of the β-NiS nanobelt was about 0.277 nm. The SAED configuration of the β-NiS nanobelt for the diffraction spots has been indexed to the planes (131), (300), and (211), as illustrated in Figure 13c. Upon the HR-TEM analysis, the β-NiS nanobelts were observed that were confirmed by the JCPDS card (120041) again. After completing the electrochemical analysis for the β-NiS nanobelt film, this specimen was analyzed by using HR-TEM analysis again. Figure 13d reveals the HR-TEM image about a nanobelt when it can be completed a long-term storage test and the electrochemical analysis from the β-NiS nanobelt film. The image also provided the certification for the β-NiS nanobelt that underwent a long-term storage test after finishing an electrochemical analysis.

## 4. Conclusions

In this work, we show that a lighter sulfide sheet combined with the nickel film could increase the chances of creating the β-NiS nanobelts. By manipulating the weight of the sulfur sheet with the nickel, film was combined at 500 °C for 4 h in a small glass tube. It can be formed into a β-NiS nanobelt film. The β-NiS nanobelts were measured at an average length of 1.5 μm. Following the long-term storage test over five years, the specimens were measured by electrochemical and optical instruments. The energy gap of the β-NiS nanobelts was measured near the value of 1.2 eV. The results of the electrochemical measurement showed that the β-NiS nanobelt film was used to detect the glucose more than 20 times in one day. The detection range of the glucose was from 1 µM to 35 µM, with a LOD of approximately 0.173 µM. It could be concluded that the β-NiS nanobelt film has the potential to develop a glucose sensor.

## Figures and Tables

**Figure 1 nanomaterials-13-02371-f001:**
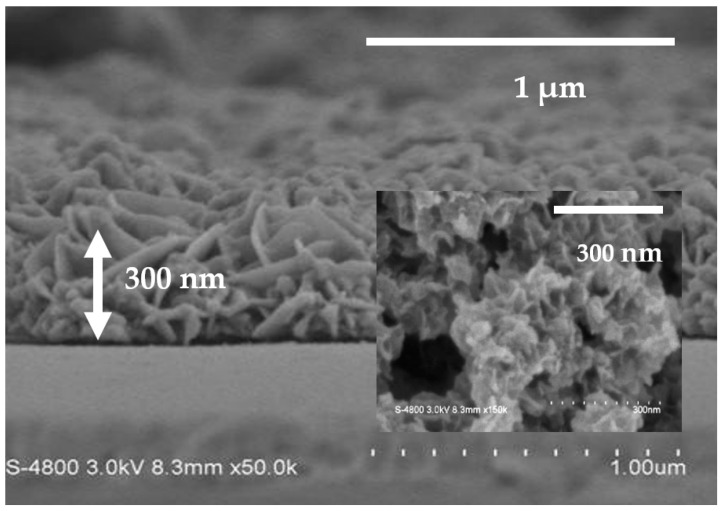
The images of FE-SEM show the cross-sectional view and the inset figure for the top view.

**Figure 2 nanomaterials-13-02371-f002:**
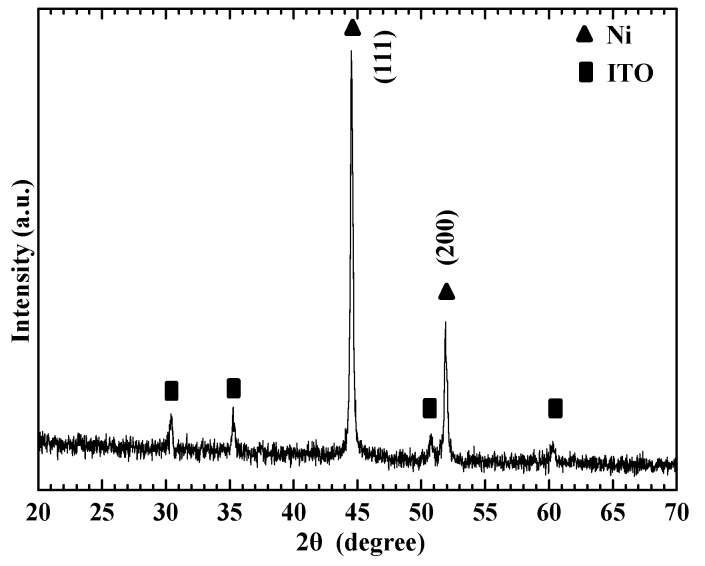
XRD pattern of the nickel film.

**Figure 3 nanomaterials-13-02371-f003:**
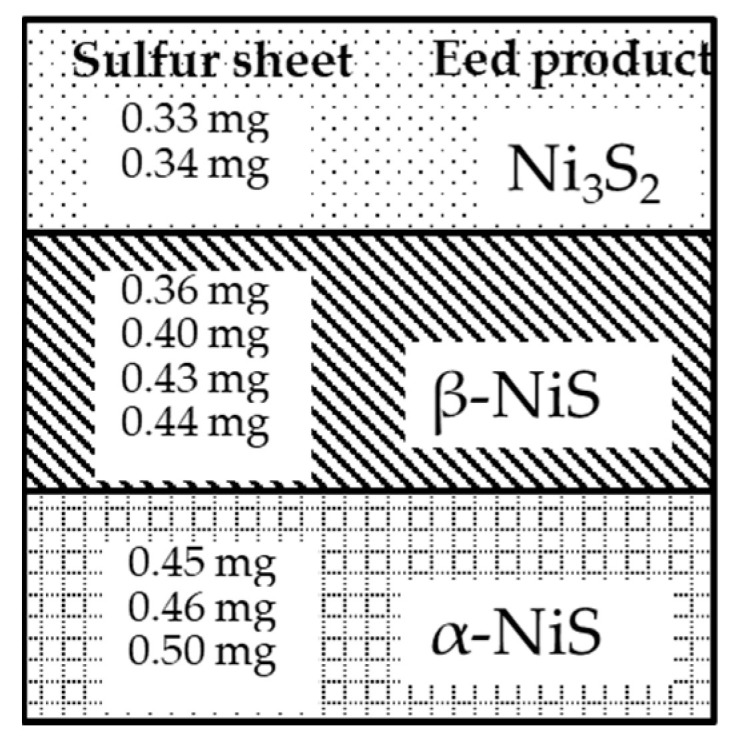
The sulfur sheets of different weights illustrate the distribution weight with the Ni_3_S_2_, β-NiS, and α-NiS films.

**Figure 4 nanomaterials-13-02371-f004:**
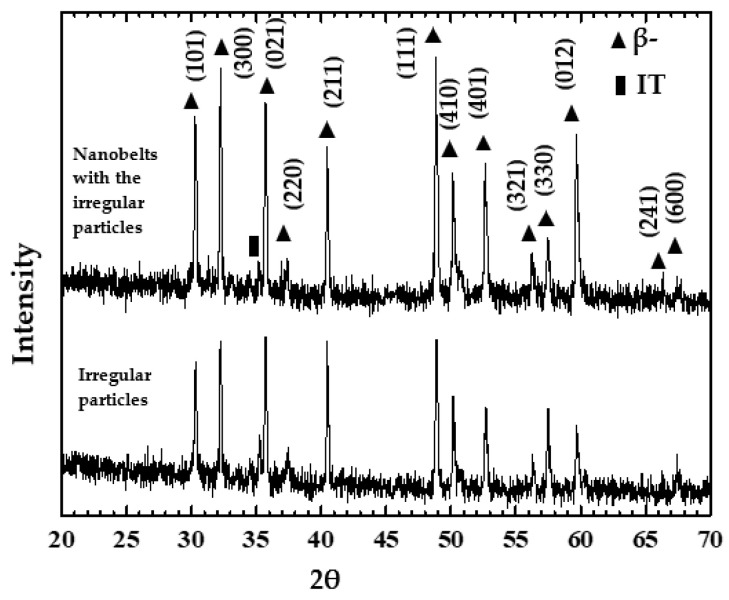
The peak patterns of the β-NiS films were the results by applying XRD.

**Figure 5 nanomaterials-13-02371-f005:**
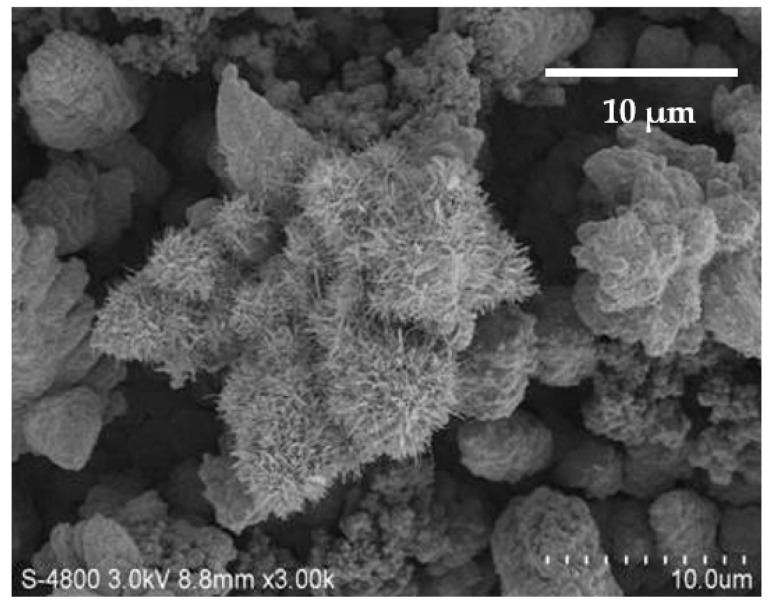
FE-SEM image of the specimen for the β-NiS film: the top view image of the β-NiS nanobelts with the irregular particles on the surface.

**Figure 6 nanomaterials-13-02371-f006:**
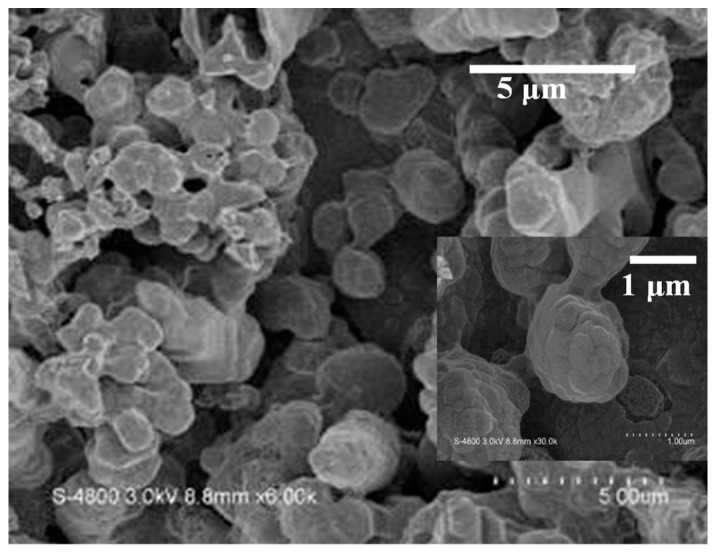
The images of the β-NiS films show the top view image of the β-NiS irregular particles by using FE-SEM.

**Figure 7 nanomaterials-13-02371-f007:**
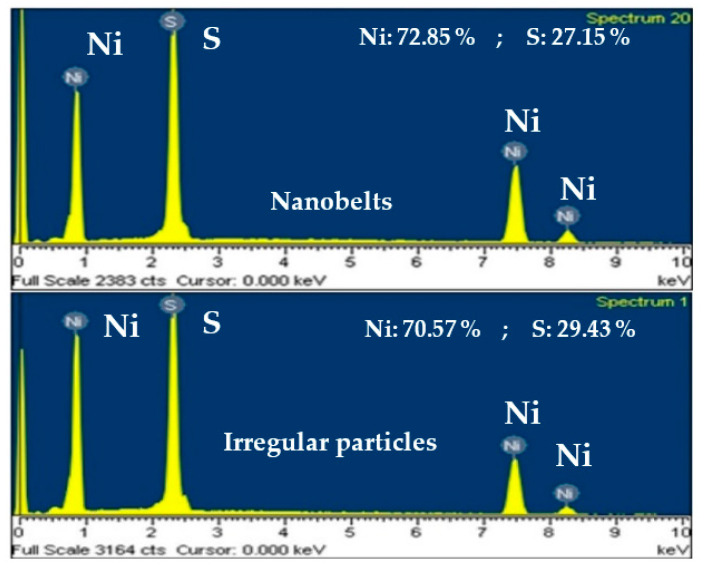
The EDS patterns of the β-NiS films.

**Figure 8 nanomaterials-13-02371-f008:**
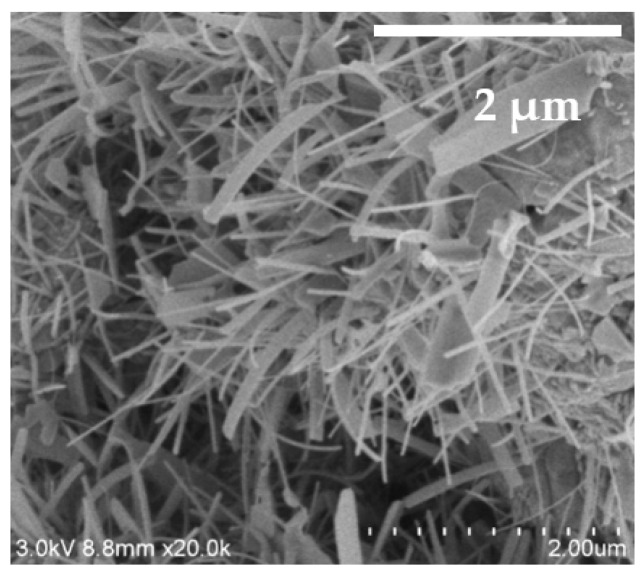
FE-SEM image of the β-NiS nanobelt film: a side image for the nanobelts.

**Figure 9 nanomaterials-13-02371-f009:**
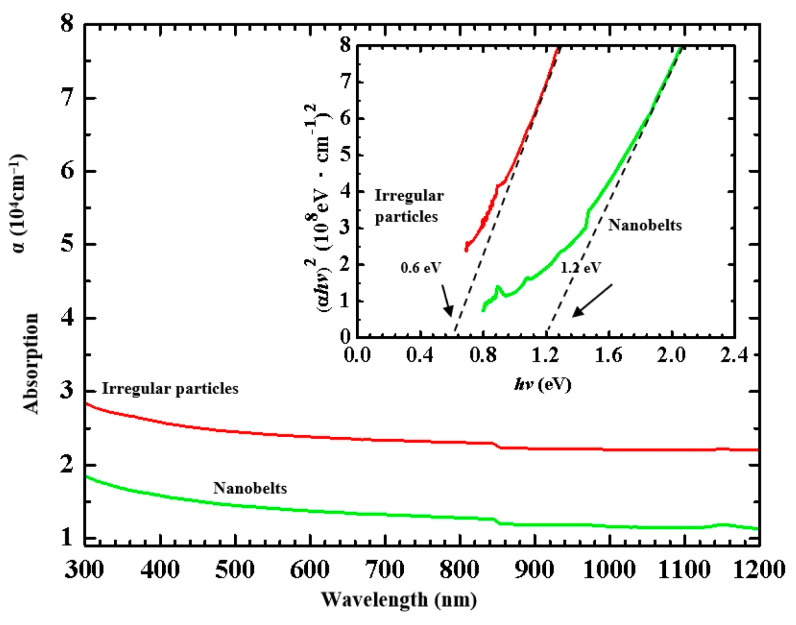
The optical absorption of the β-NiS specimens. The inset: the band gap energy of the β-NiS specimens.

**Figure 10 nanomaterials-13-02371-f010:**
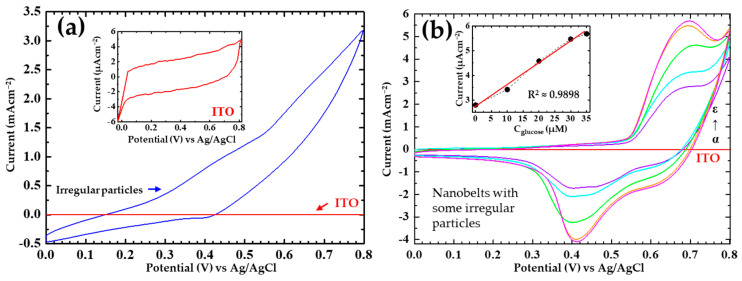
(**a**) The measurement of cyclic voltammogram shows the result of β-NiS irregular particle film in a NaOH solution. Inset image: the cyclic voltammogram shows the value of the bare ITO glass substrate. (**b**) The cyclic voltammograms (CVs) of NiS nanobelt film show the results by applying CV in a NaOH solution at variable glucose concentrations: 0 µM, 10 µM, 20 µM, 30 µM, and 35 µM. Inset: drawing of oxidation peak current against variable glucose concentrations.

**Figure 11 nanomaterials-13-02371-f011:**
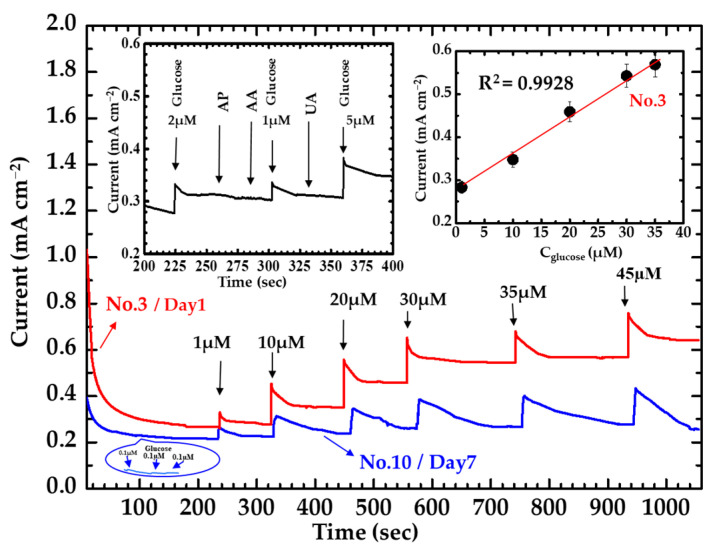
Amperometry measurement in 0.1 M NaOH with variable glucose concentrations of 1–45 µM are shows on the β-NiS nanobelt film. Inset: top right: graph of actual current responses to glucose concentrations. Inset: bottom: chronoamperometric response of β-NiS nanobelt film in 0.1 M NaOH with the three times 0.1 µM glucose. Inset: top left: chronoamperometric response of β-NiS nanobelt film in 0.1 M NaOH with the three times 2 µM, 1 µM, 5 µM glucose, and in the presence of 2 µM uric acid (UA), ascorbic acid (AA), and 4-acetamidophenol (AP).

**Figure 12 nanomaterials-13-02371-f012:**
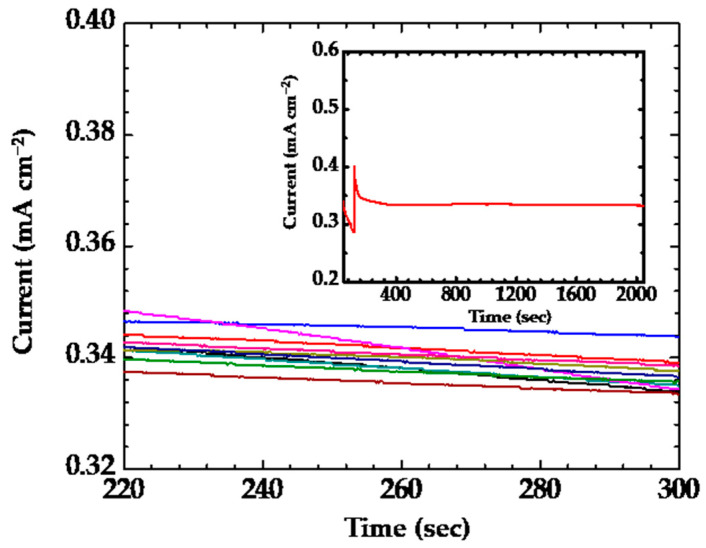
The measurement values were recorded using the ten different color lines. Ten successive amperometric curves of 10 µM glucose using β-NiS nanobelt film in 0.1 M NaOH. Inset: the stability curve of the specimen decreased around 2.6% after 30 min.

**Figure 13 nanomaterials-13-02371-f013:**
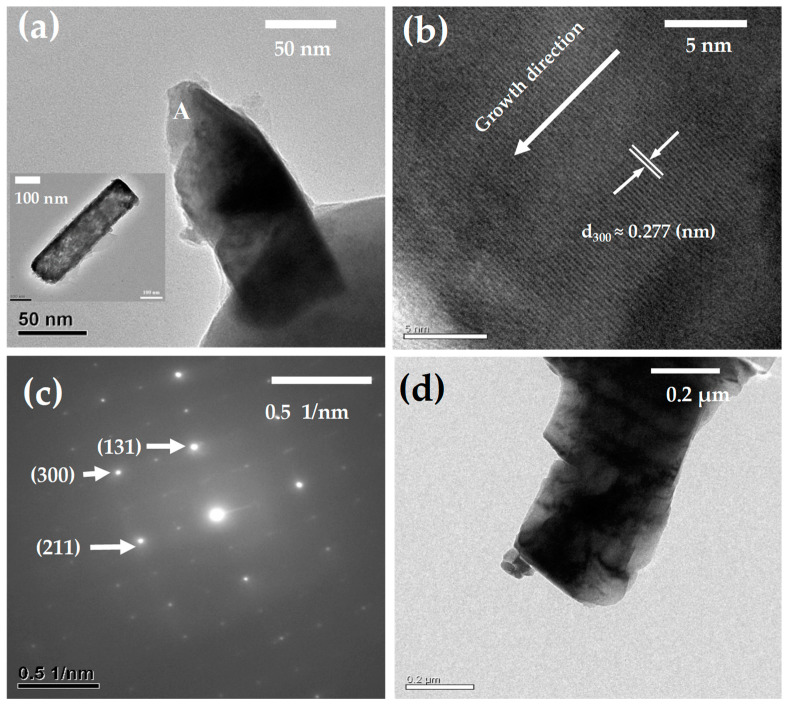
The analysis of the β-NiS specimens: (**a**) HR-TEM images for the β-NiS nanobelt. (**b**) Lattice fringes for β-NiS nanobelt. (**c**) SAED pattern of the β-NiS nanobelt. (**d**) HR-TEM image of the β-NiS nanobelt after finishing a long-term storage and the electrochemical test.

**Table 1 nanomaterials-13-02371-t001:** The results of the NiS films were obtained by annealing the different weights of sulfur sheets with the Ni film.

Number	Weight of Sulfur Sheet (mg)	XRD Analysis JCPDS Card	Final Product	FE-SEM Analysis for the Different Morphologies
No. 1	0.43	120041	β-NiS	irregular particles
No. 2	0.43	120041	β-NiS	irregular particles
No. 3	0.36	120041	β-NiS	Nanobelts + irregular particles
No. 4	0.36	120041	β-NiS	Nanobelts + irregular particles
No. 5	0.43	120041	β-NiS	irregular particles
No. 6	0.44	120041	β-NiS	irregular particles
No. 7	0.44	120041	β-NiS	irregular particles
No. 8	0.34	441418	Ni_3_S_2_	irregular particles
No. 9	0.33	441418	Ni_3_S_2_	irregular particles
No. 10	0.36	120041	β-NiS	Nanobelts + irregular particles
No. 11	0.50	750613	α-NiS	Nanoparticles
No. 12	0.45	750613	α-NiS	irregular particles
No. 13	0.46	750613	α-NiS	irregular particles
No. 14	0.40	120041	β-NiS	irregular particles

**Table 2 nanomaterials-13-02371-t002:** Compute the values of the β-NiS nanobelt films for 10 µM glucose detection for standard deviation (SD) and reproducibility (Specimen number: N = 2).

No./Day	Number of Trials/Testing Time for One Trial (Minute)	Average for the Current Response (mA)	Standard Deviation (SD)	Relative Standard Deviation (RSD)	Average for the RSD Value	Reproducibility / Error Value
No. 3/Day 1	10/5 min	0.3396	0.0028	0.82%	0.91%	99.66% / 0.34%
	10/5 min	0.3411	0.0034	0.99%		
No. 3/Day 7	2/5 min	0.3382		1.25%		
No. 10/Day 1	10/5 min	0.3311	0.0052	1.57%	1.79%	98.88% / 1.12%
	10/5 min	0.3362	0.0068	2.02%		
No. 10/Day 7	2/5 min	0.0810		3.49%		
No. 3/Day 1	1/30 min	0.3382	0.0078	2.48%		
No. 10/Day 1	1/30 min	0.3181				
No. 3/Day 1	10/0.8 min	0.3523	0.0005			

**Table 3 nanomaterials-13-02371-t003:** Comparing the glucose detection performance of nickel sulfides with some reported literature.

Electrode	Sensitivity (mA µM^−1^ cm^−2^)	Linear Range (mM)	Low Detection (µM)	Reference
α-NiS and β-NiS hollow spheres	0.0036	0.000125–2.0	20	[7]
NiS	0.00743	0.005–0.045	0.32	[8]
α-NiS nanospheres	0.0084	0.001–0.035	1.0	[9]
NiS micro urchin	–	0.05–1.7	10	[10]
Ni_3_S_2_ nanosheet arrays	6.148	0.005–3.0	1.2	[38]
Ni_3_S_2_ hierarchical nano cauliflower/NF	16.46	0.0005–3.0	0.82	[39]
NiS/S-g-C_3_N_4_ nanoparticles	0.08	0.001–2.1	1.5	[41]
NC-NiS@NS-NiS nanoparticles	0.0546	0.02–5.0	0.0083	[42]
β-NiS nanoparticles	0.00578	0.005–0.06	0.052	[43]
β-NiS nanobelts	0.00867	0.001–0.035	0.381	This work

## Data Availability

The data presented in this study are available on request from the corresponding authors.

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
