# Peer review of "The Slight Adjustment in the Weight of Sulfur Sheets to Synthesize β-NiS Nanobelts for Maintaining Detection of Lower Concentrations of Glucose through a Long-Term Storage Test"

_nanomaterials, 2023, doi:10.3390/nano13162371_

Round 1

Reviewer 1 Report

The manuscript deals with the fabrication of β-NiS nanobelts and their application as a sensing layer for the electrochemical detection of glucose. The topic is well-studied to date and there is a lack of scientific novelty in the research presented. The method of β-NiS nanomaterials is also already reported before and similar approach has been used to get β-NiS nanobelts.

Another important point is the reproducibility of the β-NiS nanobelts production. As follows from the  text, there was just one sample of the nanomaterial that has been tested. Glucose test has been performed after 5 years of storage. The initial response parameters of glucose obtained after preparation of the nanomaterial have to be also tested for the adequate comparison.

The application to glucose detection of such type of electrodes is also well-known as Authors summarized in Table 2. Furthremore, the analytical characteristics obtained are questionable as far as linear regression presented in Figure 14 has low level of linearity. The R2 should be at least 0.999 for the quantitation purposes. There are another important drawbacks in the manuscript.

1. English needs revision. Please, remove in almost each phrase "We". Scientific style implies an impersonal form of presentation.

2. Introduction is too short and does not show the state-of-the-art of the topic.

3. Figures 1-3 are not meaningful and can be easily removed.

4. Experimental section is incomplete. Conditions of electrochemical measurements are required.

5. It is unclear from the text, which sample is shown in Figure 11.

6. Analytical section has serious flaws from methodological point of view.

a) All data are preseneted as a single measurement results that is imappropriate. The error bars and SD values are lost.

b) Linearizations are not shown. Slopes and intercepts presented  are questionnable as far as number of significant digits is wrong taking into account the plots presented.

c) The accuracy, repeatability, reproducibility, stability and selectivity tests are fully out of consideration.

d) How was the limit of detection calculated?

7. Table 2 is confusing. The same units have to be shown for all data.

Thus, summarizing mentioned above, I can not recommned the manuscript to publication.

English needs revision in style. Scientific style implies an impersonal form of presentation. The current form decreases manuscript readability.

Author Response

Hi Reviwer 1,

Please kindly check the attached file. Thanks!!

Reviewer 2 Report

Authors propose novel non-enzymatic glucose sensors, which are β-NiS nanobelts on ITO glass substrates. Authors have performed a thorough investigation of the obtained electrodes, including morphological analysis, electrochemical properties, XRD analysis. The obtained sensors demonstrate high sensitivity for glucose with a wide linear range. Importantly, the authors have performed a very long-range test checking the stability of the sensors, and also provided a thorough comparison with other glucose sensors. The article is well-written, easy-to-read and contains all necessary illustrations. I think the manuscript is acceptable for publication after a major revision.

Major revision. Authors should provide results on the selectivity of sensors. The selectivity can be tested on some real in vitro blood samples or by performing measurements in the presence of interfering species such as ascorbic acid (AA), uric acid (UA), 4-acetamidophenol (AP).

Author Response

Hi Reviewer 2,
Please kindly check attached file. Thanks!!

Reviewer 3 Report

The manuscript describes synthesis of β-NiS nanobelts as active film for enzymeless glucose detection in alkaline solution. The idea is not new, however, the authors synthesised different shape of nanoparticles which extended life-time of the sensor. Manuscript needs major revision.

1. Introduction and conclusions need to be restructured. Now, when the information is in one paragraph, it is too confusing.

2. Common English expressions have to be used in the manuscript, e.g. limit of detection.

3. Calibration plots need total revision. Please see in other papers how to present calibration plots and what parameters should be taken from it. Now the results are not reliable seeing such a plots.

4. Why TEM characterisation is presented after application of the sensor? Is a particular reason for it? If not, please restructure the manuscript. If yes, please explain it in the text.

5. Figure 14: Why so big of the signal is observed? Which part of the response was taken to make a calibration plot? It is better to present calibration plot separately as part B and rescale the chronoamperogram to have a clearer view of the response.

6. Table 2. Please add the methods of the signal acquisition because the method has also impact on the analytical parameters. Please recalculate sensitivity to the same units. The authors are not obliged to give values in originally given units. If the linear range from [36] is given in mM, why the linear range obtained by the authors is given in micro mols per liter? It needs unification.

7. No information about stability (except conclusions) and reproducibility as well as reliability of such sensor is given.

8. The authors claim that they can detect low glucose concentrations. Please specify then what kind of natural samples could be used for glucose analysis with this sensor. Blood has much higher concentrations as well as it has physiological pH, however, this sensor operates only in alkaline medium.

English needs just minor revision.

Author Response

Hi Reviewer3,
Please kindly check the attached file. Thanks!!

Round 2

Reviewer 1 Report

The revised manuscript is improved vs. initial submission. Nevertheless, it still has points to be corrected and clarified.

1. Abbreviations need revision. All of them used 1-2 times throughout the manuscript to be removed.

2. Linear plots (inserts in Figures 10b and 11), the linear fit has to be applied to the points on the plot. The straight line that just connects points is shown in current version. Furthermore, replace the X-axis title to cglucose (μM). The molar concentration of glucose is measured.

3. The exact value of R2 with four decimal places accuracy to be presented throughout the manuscript including text and figures.

4. Equation 4, the slope and intercept to be presented as average value±SD taking into account the number of decimal places. Moreover, replace [glucose] μM with cglucose [μM]. Concentration marked as squiare brackets means equilibrium concentration that is inapplicable in electrochemical approaches used. The molar concentration is used.

5. Lines 302 and 303, the text and data are shown in incorrect way. Accuracy of the sensor can not be presented by the value of current. Text to be rephrased. The secon point is  0.3405mA±0.911% (No.3) or 0.3335mA±1.795% (No.10). Similar units to be used for the presentation, i.e. 0.341±0.003 mA and 0.334±0.006 mA, where 0.003 and 0.006 are the currents calculated as 0.911 and 1.795% from average current values.

6. Line 304, the reproducibility of 0.212% (No.3) and 2.12% (No.10) means the absence of reproducibility. The standard deviation or the error values in the reproducibility test are 0.212% (No.3) and 2.12% (No.10) that means 99.8 and 98% of data reproducibility. Please, use correct terms for the data. Otherwise, You confuse the readers.

7. Lines 306-307, please, think of the meaning of "The average current response is the value of 0.314 mA during the testing time up to 30 min. The number of the deviation is the value of Δ26 mA." This is impossible to understand.

8. Fig. 12 caption should be revised as "Ten successive amperometric curves of 10 μM glucose using the β-NiS nanobelt film in 0.1 M NaOH".

9. Table 2. Replace abbreviation RDS to RSD. Furthermore, last column "Reproducibility" rename as "RSD in reproducibility test". The number of measurements should be presented for data in Table 2.

10. Line 321,"The Sb is the standard deviation". Explanation of standard deviation is needed. Which data were used for it?

11. Table 3, the LOD of 0.341μM should be presented for the current work as mentioned in line 322.

English style needs revision.

Author Response

Hi Reviewer,
We have revised the manuscript. Please kindly check the attached file. Thank you!!

BR

Hsien-Sheng Lin

Reviewer 2 Report

In the revised version of the manuscript the authors have added results of experiments considering the selectivity of sensors. The manuscript can be accepted in present form.

Author Response

Hi Reviewer,
We have revised the manuscript. Please kindly check the attached file. Thank you!!

B.R.

Hsien-Sheng Lin

Reviewer 3 Report

The authors probably have submitted the wrong file:

1. There are no marked changes.

2. Plots are not corrected.

One more comment: Fig. 14 caption has Greek letters but they are not presented on the plot. It is recommended to omit the concentrations from caption because everything is described on plot.

Author Response

Hi reviewer,
We have revised the manuscript. Please kindly check the file. Thank you!!

BR

Hsien-sheng Lin

Round 3

Reviewer 1 Report

The manuscript is improved a bit but several points become even more unclear. The answers to the comments are not meaningful.

The manuscript still needs significant revision.

1. English needs corrections in style and grammar (check carefully the modified paragraphs).

2. Comment 4 to previous version. Equation 4, the slope and intercept to be presented as average value±SD taking into account the number of decimal places i.e

I [mAcm-2] = (8.67 ± 0.76) × Cglucose [μM] + (0.28 ± 0.03).

In this case, the text on lines 318-319 and lines 321-323 (marked with yellow) should be removed to avoid the repetition.

Furthermore, line 313 the sensitivity value 8.8 μA cm-2 μM-1 should be corrected as 8.67 ± 0.76 μA cm-2 μM-1.

Line 312, replace correlation coefficient of 0.99 with 0.9928.

3. Sentence on the lines 310-311 to be removed due to the repetition of the information presented in line 312.

4. Lines 349-353, the data are inconsistent and confusing. Furthermore, the RSD values are usually presented with two significant digits.

The text "From the values of the average current response and relative standard deviation (RSD) in the Table 2, it can be observed that the values of the current response with the deviation are the average RSD value of 0.9104%(No.3) or 1.7945%(No.10). These values of the deviation are smaller (less than 2%). It can be also accepted that the absence of reproducibility are the values of 0.3394% (No.3) and 1.1166%(No.10). The reproducibility of the specimens are the values of 99.6606%(No.3) and 98.8834%(No.10). The results are satisfied the general measurement system (less than 10%) [44]." should be rephrased as:

"The average RSD values of 0.91% (No.3) or 1.79% (No.10) being significantly less than 10% and the reproducibility of 99.66% (No.3) and 98.88% (No.10) satisfy the general measurement system quality [44]."

5. Lines 354-360, "About the stability of the measurement was carried out at 10 μM glucose by using specimens (No.3) over 30 min. From inset of Figure 12, the stability curve of the specimen decreased around 2.6% after 30 min. The value of the average current response is 0.3381 mA and final value of the measurement is 0.3293 mA after testing 30 min. The stability of the measurement for the deviation is the value of 2.6% under 10% of the average current response. The specimen (No.3) could be believed that numerical value of stability is good enough for sensing the glucose."

replace with

"The stability of the measurement was carried out at 10 μM glucose by using specimens (No.3) over 30 min. The curent response of the specimen decreased around 2.6% (from 0.3381 to 0.3293 mA) after 30 min (Figure 12, insert). The deviation obtained is less than 10% of the average current response. Thus, the specimen (No.3) shows enough stability of response for  glucose sensing."

6. Table 2, data for RSD, average RSD and Reproducibility/error value (last 3 columns) should be presented with two decimal places, i.e. 0.82%, 0.91%, and 99.66/0.34%, etc.

7. Comment 10 to previsous version (Line 373 in current manuscript), I suggest that Sb is the standard deviation of the currents for blank solution in 10 parallel measurements. Is this right?

8. LOD and LOQ have to be recalculated using the average value of the calibration graph slope which is 8.67 μA cm-2 μM-1 but not 8.8 μA cm-2 μM-1. The corresponding corrections have to be inserted throughout the manuscript including the abstract, manuscript text, table 3 and conclusions.

9. Conclusions need revision as far as repeats the abstract that is unacceptable.

English style and grammar needs revision especially in the revised paragraphs.

Author Response

Hi Reviewer1,
Please kindly check the attached file. 

B.R.

Hsien-Sheng Lin

Reviewer 3 Report

The manuscript now has much better quality. Only the comment No. 2 is not considered in the text although the answer says that everything is corrected. Please, use common terms of Analytical Chemistry.

Common expressions of analytical chemistry should be used. "Low detection limit" has to be replaced with "limit of detection" or LOD.

Author Response

Hi Reviewer,

Please kindly check the attached file. Thank you!!

B.R.

Hsien-Sheng Lin
